# Patients' perceptions of care in a Type 1 hybrid effectiveness-implementation trial on intravenous iron for anaemia in pregnancy in Nigeria

Mobolanle Balogun[1,2*], Opeyemi R. Akinajo[2,3], Rachel A. Thompson[2], Teniola Lawanson[2], Hameed Adelabu[2], Nadia A. Sam-Agudu[4,5,6], Bosede B. Afolabi[2,3]

1 Department of Community Health & Primary Care, College of Medicine, University of Lagos, Lagos, Nigeria, 2 Centre for Clinical Trials, Research and Implementation Science, College of Medicine, University of Lagos, Lagos, Nigeria, 3 Department of Obstetrics & Gynaecology, College of Medicine, University of Lagos, Lagos, Nigeria, 4 International Research Center of Excellence, Institute of Human Virology Nigeria, Abuja, Nigeria, 5 Department of Paediatrics and Child Health, University of Cape Coast School of Medical Sciences, Cape Coast, Ghana, 6 Global Pediatrics Program and Division of Infectious Diseases, Department of Pediatrics, University of Minnesota Medical School, Minneapolis, United States of America

* mbalogun@cmul.edu.ng

## Abstract

### Introduction

Iron deficiency anaemia is a common cause of anaemia in pregnancy, which is highly prevalent in low-and middle-income countries (LMICs). Evidence from LMICs indicates that intravenous (IV) iron formulations such as ferric carboxymaltose (FCM) are safe and effective alternatives to oral iron, and there is emerging data on implementation strategies and outcomes. Client outcomes are important additional considerations for successful scale-up of new interventions. We assessed patient experiences and satisfaction among women receiving IV versus oral iron for anaemia in pregnancy in Nigeria.

### Methods

This mixed-methods study was nested in a type 1 hybrid effectiveness-implementation trial that enrolled 1,056 pregnant women at 20–32 weeks gestational age. Participants were randomised to receive either FCM or oral ferrous sulphate (FS). Twenty-five percent of participants were sequentially sampled for interviewer-administered exit surveys and 66 women who received IV iron were purposively selected for in-depth interviews at last study visit. Quantitative data from the two treatment groups were evaluated using the Chi-squared test, while qualitative data were analysed thematically.

**Data availability statement:** Data cannot be shared publicly because of potentially identifying patient information. Data are available from the Center of Clinical Trials, Research and Implementation Science Institutional Data Access (contact via info@cctris.org) for researchers who meet the criteria for access to confidential data.

**Funding:** The main trial titled "Intravenous versus oral iron for iron deficiency anaemia in pregnant Nigerian women (IVON)" is an open-label, randomised controlled trial funded by the Bill & Melinda Gates Foundation (BMGF) Grant (Investment ID INV-017271). The funders had no role in study design, data collection and analysis, decision to publish, or preparation of the manuscript.

**Competing interests:** The authors have declared that no competing interests exist.

## Results

We surveyed 252 women (128 treated with FCM and 124 with FS). Significantly higher proportions of the FCM (73.4%) versus FS group (57.3%) had positive perceptions of care (p = 0.007). Positive perceptions, experiences and satisfaction with FCM were buttressed by the qualitative findings, for reasons such as good communication and quality of provider care, single-dose administration, minimal side effects, positive health outcomes and not bearing treatment costs.

## Conclusion

Participants in this IV versus oral iron trial had positive perceptions of FCM. These patient-reported findings support available evidence on service and implementation outcomes and further support IV iron scale up in LMIC settings. Future implementation research should further assess client outcomes under real-world conditions.

## Introduction

Anaemia in pregnancy is a global health issue affecting both high and low-middle-income countries (LMICs) [1]. The most common type of anaemia in pregnancy is iron deficiency anaemia, with an estimated prevalence of 50–75% worldwide [2,3]. Evidence suggests that LMICs have significantly higher rates of anaemia in pregnancy, possibly due to economic, sociological, and health factors related to pregnancy [4]. In the African region, the estimated prevalence of anaemia in pregnancy is about 43% [5]. In Nigeria, about 20–40% of pregnant women are anaemic [6,7]. Anaemia in pregnancy can lead to several adverse maternal and foetal outcomes, including increased risk of postpartum haemorrhage and intrauterine foetal death [8].

Oral iron is the standard treatment in Nigeria for mild to moderate cases of anaemia in pregnancy [5]. However, the commonly used form of oral iron, ferrous sulphate (FS), is linked to various side effects and low adherence to treatment [7,9]. Additionally, due to poor health-seeking behaviour and financial constraints, many women fail to complete the standard oral iron regimen [10]. This non-compliance with oral iron increases the risk of undertreated and persistent anaemia in pregnancy [3,11]. Intravenous (IV) iron has proven to be a viable alternative to oral iron for the treatment of anaemia in pregnancy [12]. Previously, high-molecular-weight IV iron formulations were infrequently used in clinical practice due to their tendency to cause severe allergic reactions and anaphylaxis [7,12]. However, newer third-generation, low-molecular-weight formulations—like ferric carboxymaltose (FCM)—are both effective and safe for use during pregnancy [13].

FCM is a recent iron formulation devoid of dextran, featuring a nearly neutral pH, physiological osmolarity, and enhanced bioavailability [14]. This formulation enables a single-dose administration with a 15–20-minute infusion time and the option for higher dosages (up to 1,000 mg) [13,14]. These characteristics make FCM a compelling alternative for other iron formulations, considering factors such as risk profile, effectiveness, patient comfort, convenience, and the utilisation of staff and institutional resources [13,14].

Studies conducted in LMICs indicate that FCM is faster and safer than oral iron therapy for treating anaemia in pregnancy [12,13]. However, demonstrating the effectiveness of an intervention such as FCM, does not automatically equate to its widespread acceptance and use. Therefore, while determining effectiveness is vital, the success of implementation is equally significant. One key factor in understanding the context for implementation success is user satisfaction, which is crucial to how well this intervention is embraced and integrated within a healthcare system [15]. Since FCM is a new treatment in the Nigerian context, it is imperative to assess users' perceptions and satisfaction with IV iron compared to oral iron. Comparisons of treatment perceptions and experience between women receiving FCM and standard of care (FS) would help to better understand and address the preferences of anaemia in pregnancy patients, given that experience of care is an important indicator of quality [16].

Thus, the objectives of this study were to assess and evaluate patient satisfaction, compare the perceptions of women who received IV versus oral iron for the treatment of anaemia in pregnancy, and gain a deeper understanding of their experiences and the factors that influence the uptake of this intervention.

## Methods

### Study design

This mixed-methods study combines quantitative and qualitative data collection methods to provide a comprehensive perspective on users' perceptions, experience and satisfaction with IV iron therapy. This approach allowed for a deeper understanding of the research topic and examined how the results from each method support or contradict one another [17]. This study was conducted as a part of a hybrid type 1 effectiveness-implementation trial that compared IV and oral iron in the treatment of iron deficiency anaemia among pregnant women in Nigeria (IVON) [6,18]. We used Proctor's framework for implementation outcomes to select and assess acceptability, feasibility, fidelity and cost-effectiveness as the implementation outcomes in the trial; service outcomes assessed were effectiveness and safety [6,10]. This current study assesses the patient (client) outcome of satisfaction (Fig 1). We followed the Standards for Reporting Implementation Studies (StaRI) checklist to report our findings (S1 Checklist) [19].

### Study setting

This study was conducted in Nigeria, with a population of over 200 million people, an average of seven million births annually, and which accounts for 29% of annual maternal deaths worldwide [20,21]. Selected study states were the two most populous states in the country: Kano in the North-West and Lagos in the South-West [22]. The two states have different

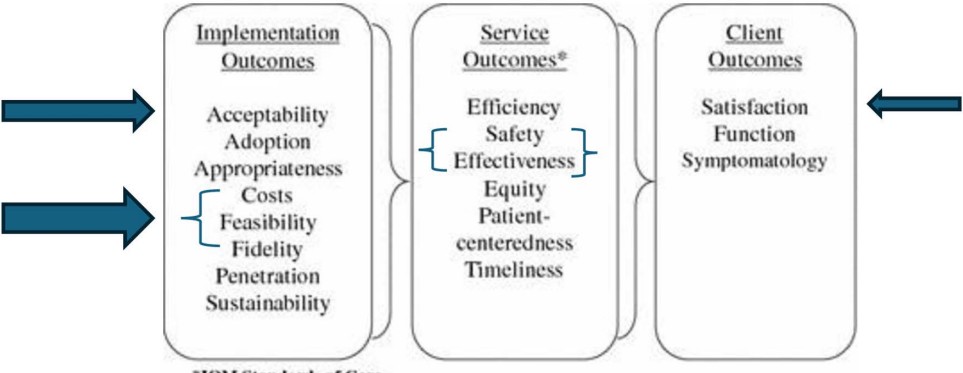

**Fig 1. Representation of Proctor's 2011 schematic with Implementation, Service and Client Outcomes relevant to the IVON Trial.**

cultures, educational levels, and healthcare service utilisation [22]. The 11 public health facilities involved in the IVON trial included two primary health centres (PHCs) in Lagos, three PHCs in Kano, and two secondary and one tertiary facility in each state. The IVON trial randomly assigned eligible pregnant women between 15 and 49 years old and 20–32 weeks gestational age into the oral FS (control) group or IV FCM (intervention) group. Participants in the FCM group received a single infusion over 15–20 minutes while participants in the FS group received 200 mg tablets three times daily until 6 weeks after delivery. A total of 1,056 women participated in the trial, with 527 in the FCM group and 529 in the oral iron group, all of whom provided written informed consent [18].

### Participants recruitment and sampling technique

**Quantitative phase.** This study involved postpartum women who had received FCM or FS between 20–32 weeks gestational age who were followed up from enrollment, with regular checks every four weeks until 28 weeks' gestation, biweekly until 36 weeks' gestation, and weekly until delivery. Additionally, these women were followed up at two-week and six-week postpartum clinic visits following delivery. Participants were eligible for an exit survey at their scheduled postnatal clinic visit at six weeks postpartum, the timepoint IVON participant follow-up was completed and they exited the trial. We aimed to survey 25% of women from each study arm, using systematic random sampling programmed into the data collection platform, to select every fourth participant, resulting in a total of 132 participants in each group and 264 in total. The contribution of each facility was adjusted based on the total number of participants enrolled at that site.

**Qualitative phase.** We purposively selected participants who had IV iron therapy based on various characteristics such as age, parity, prior experience with oral iron therapy and those who receive care from healthcare facilities known for their high patient volumes. We also considered unique circumstances surrounding the administration of IV iron therapy such as the length of waiting time for receiving IV iron, any apprehensions such as needle pricks, heightened anxiety related to the administration process and occurrence of side effects if any. We used the concept of information power to ensure that our gathered data was both sufficient and relevant [23]. This approach considers the study's specific objectives, the unique characteristics of the sample, and the valuable, high-quality insights provided by the users, allowing us to gain a comprehensive understanding of their experiences with the intervention [23].

### Data collection tools

For the exit survey, the research team designed a 10-item questionnaire to collect data on participant's perceptions of care and satisfaction regarding the treatment (FCM or FS) they received during the trial. The survey assessed trial participants' perceptions about the communication from providers about their treatment; the challenges experienced during the treatment, including side effects; overall satisfaction with information and support received from providers; and recommendation of treatment to a family member. The questions had Likert scale responses that were relevant for each question, e.g., never, a few times, most of the time and all the time; and excellent, good, and fair. Before data collection, the survey underwent a rigorous review process by the research team, was corrected, and was piloted with non-participating pregnant women to ensure the validity and reliability of the tool.

Qualitative data were collected through in-depth interviews (IDIs) with the purposively selected participants. A semi-structured topic guide with five sections was developed to explore users' knowledge of anaemia in pregnancy, their perceptions of and relative advantages of IV iron compared to oral iron, and their challenges and experiences with the administration process, including overall satisfaction with IV iron therapy.

### Data collection methods

All sampled participants were contacted before their six-week postpartum visit to inform them about the exit survey and the IDIs and to ascertain their interest in participating. Of the 264 participants contacted across all the healthcare facilities, 260 were reachable, and 252 agreed to participate in the exit surveys (Fig 2). On the day of their postpartum visit,

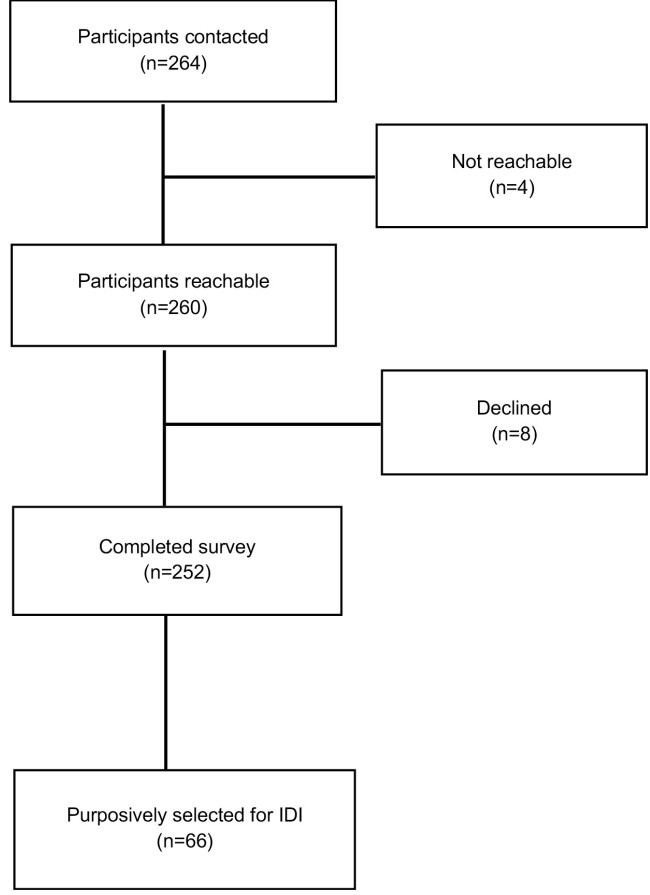

**Fig 2. Flow chart showing the number of participants involved in the study.**

participants who agreed to participate were interviewed after informed consent was obtained. Eight trained research assistants who were independent of the clinical trial administered the surveys. To reduce social desirability bias, the 30–45-minute surveys were conducted in a private room at the health facility after each consenting participant was discharged from the trial by the research staff.

For the IDIs, we piloted the interview guide before data collection and made necessary adjustments based on the participants' responses. We chose IDIs as they provided a practical opportunity for one-on-one discussions with the participants who received IV iron therapy [24]. In total, we interviewed 66 participants concurrent with their exit surveys; participants were purposively selected from all 11 IVON study facilities (Fig 2). All IDIs were conducted by the same research assistants who administered the exit surveys. Each interview lasted between 15 and 40 minutes. Participants' responses were read back to them for accuracy and clarity of intended meaning. All survey and interview participants were compensated with transportation fare.

Data were collected between October 2021 and January 2023. Quantitative data were collected using an electronic device (tablet) and entered directly into Research Electronic Data Capture (REDCap) [25,26]. For the qualitative data, we obtained verbal consent from each participant before commencing the audio recordings. Subsequently, each interview was audio-recorded, and all interviews were transcribed verbatim.

### Data analysis

**Quantitative data analysis.**  STATA version SE15.1 (StataCorp, College Station, Texas, USA) was used to analyze the data. The continuous variable, age, was tested for normality using the Shapiro-Wilk test. Descriptive statistics such as frequencies, percentages, and median and interquartile range, were used to describe sociodemographic characteristics. Differences in categorical variables between the two groups were evaluated using the Chi-squared test.

Perception was quantified by summing nine recoded items reflecting provider interactions, treatment experience, and overall evaluation. Five items were scored on a similar four-level scale (0 = "No, never" to 3 = "Yes, all the time"): felt provider explained what was done; provider spoke a language understood; provider explained why he/she was giving medicine; participant felt she could ask questions, and participant felt provider gave their best care. Ease of treatment administration was scored from 1 = "Extremely difficult" to 5 = "Easy; no challenge", and perceived side effects were scored from 1 = "Extremely difficult" to 5 = "No side effects". Overall satisfaction was scored from 1 = "Very poor" to 5 = "Excellent", while willingness to recommend treatment was scored from 1 = "Very strongly do not rec-ommend" to 7 = "Very strongly recommend". The lowest to highest attainable total perception score was 0–37. The overall median score was then used as the cut-off to classify perception as positive (score ≥ the median) and negative (score < median); we used the same threshold for both treatment groups (FCM and FS) to enable direct comparability. Sociodemographic factors associated with overall perception of care were evaluated using Chi-squared or Fisher's exact test. The level of significance was set at 0.05.

**Qualitative data analysis.**  All transcribed interviews were analysed using a manual rapid thematic analysis approach. Two qualitatively trained researchers (MB and RAT) coded a subset of the transcripts independently using a deductive approach to create a codebook, which was applied to the rest of the dataset. Next, we developed, reviewed, and refined our themes to accurately reflect the dataset's meaning.

**Integration of Quantitative and Qualitative Data.**  In this study, we integrated quantitative and qualitative methods of data collection, focusing on participants who received IV iron, a new intervention with a distinctly different mode of administration compared to oral iron. Here, we considered both data collection methods complementary and of equal significance in our research design. To effectively integrate findings, we employed a joint display matrix which enabled us to systematically compare and contrast survey versus interview findings, highlighting the relationships between the numerical data and the contextual information derived from qualitative data [27].

### Ethical considerations

Ethical approval was obtained from the National Health Research Ethics Committee of Nigeria (NHREC/01/01/2007–17/01/2021), Health Research and Ethics Committees of the Lagos University Teaching Hospital (ADM/DCST/HREC/APP/3971), Aminu Kano Teaching Hospital, Kano State (NHREC/28/01/2020/AKTH/EC/2955) and Ministry of Health, Kano State (MOH/Off/797/T.1/2102). Permission was obtained from the Lagos State Health Service Commissions (LSHSC/2222/VOLIII) and the Lagos State Primary Health Care Board (LS/PHCB/MS/1128/VOL.VII/100) for using the sites in Lagos state. As this study was considered minimal risk, we obtained verbal informed consent from all participants before data collection.

### Results

#### Quantitative results

A total of 252 women participated in the survey; 128 had been treated with FCM while 124 had been treated with FS. Median age was 27 years (IQR: 22–32) with age range between 16 and 44 years. Most participants were married (96.2%), attended secondary health facilities, and resided in urban areas (94%); 49.6% had received secondary education and 44.4% were self-employed (Table 1).

**Table 1. Sociodemographic characteristics of respondents.**

| Variables | FCM group (N=128) | FS group (N=124) | Total (N=252) |
|---|---|---|---|
| | n (%) or Median (IQR) | | |
| **Age, years** | 27.5 (17-39) | 27 (16-44) | 27 (22–31) |
| **Marital status** | | | |
| Single | 6 (4.7) | 1 (0.8) | 7 (2.8) |
| Cohabiting | 2 (1.6) | 2 (1.6) | 4 (1.6) |
| Married | 119 (93.0) | 121 (98.0) | 240 (96.2) |
| Divorced | 1 (0.8) | 0 (0.0) | 1 (0.4) |
| **Education** | | | |
| No formal education | 9 (7.0) | 5 (4.0) | 14 (5.6) |
| Completed primary | 13 (10.0) | 19 (15.0) | 32 (12.7) |
| Completed secondary | 64 (50.0) | 61 (49.0) | 125 (49.6) |
| Completed tertiary | 39 (30.0) | 36 (29.0) | 75 (29.8) |
| Postgraduate | 3 (2.3) | 3 (2.4) | 6 (2.4) |
| **Place of residence** | | | |
| Rural | 8 (6.2) | 7 (5.6) | 15 (6.0) |
| Urban | 120 (94.0) | 117 (94.0) | 237 (94.0) |
| **Ethnicity** | | | |
| Hausa | 64 (50.0) | 62 (50.0) | 126 (50.0) |
| Igbo | 14 (11.0) | 15 (12.0) | 29 (11.5) |
| Yoruba | 41 (32.0) | 37 (30.0) | 78 (31.0) |
| Others | 9 (7.0) | 10 (8.1) | 19 (7.5) |
| **Occupation** | | | |
| Unemployed | 51 (40.0) | 49 (40.0) | 100 (39.7) |
| Self-employed | 62 (48.0) | 50 (40.0) | 112 (44.4) |
| Private employment | 9 (7.0) | 15 (12.0) | 24 (9.5) |
| Government employment | 4 (3.1) | 8 (6.5) | 12 (4.8) |
| Unknown | 1 (0.8) | 0 (0.0) | 1 (0.4) |
| Not applicable | 1 (0.8) | 2 (1.6) | 3 (1.2) |
| **State** | | | |
| Kano | 67 (52.3) | 65 (52.4) | 132 (52.4) |
| Lagos | 61 (47.7) | 59 (47.6) | 120 (47.6) |
| **Facility type** | | | |
| Primary | 39 (30.5) | 39 (31.5) | 78 (31.0) |
| Secondary | 77 (60.2) | 75 (60.5) | 152 (60.3) |
| Tertiary | 12 (9.4) | 10 (8.1) | 22 (8.7) |

In comparing their perceptions of care, most women in the FCM and FS groups felt their providers explained treatment procedures "all the time" (85.9% and 75.8% respectively) and felt providers spoke in a language they could understand "all the time" (92.2% and 91.9% respectively). Most respondents also felt providers explained why they gave medicine "all the time" (90.6% and 92.7% respectively), felt they could ask any question "all the time" (75.6% and 73.4% respectively), and felt the providers took the best care of them "all the time" (91.4% and 93.6% respectively). Almost all the respondents (96.1% and 91.9% respectively) denied challenges with side effects. These perceptions were not statistically different among the two groups (p>0.05).

A significantly higher proportion of respondents receiving FCM (82.8%) perceived their treatment to be "easy" compared to 63.7% receiving FS (p = 0.006) (Table 2). When asked about overall satisfaction with information and support received from providers, a significantly higher proportion of respondents in the FCM group (80.5%) rated it as "excellent" compared to 62.1% in the FS group (p = 0.005). Significantly more people in the FCM vs the FS group (70.3 vs 50.0%) would also "very strongly recommend" the treatment to a family member (p = 0.018). Total perception scores ranged from 25 to 37 out of a maximum of 37 with a median value of 35 (IQR: 34–36). Overall, a significantly higher proportion of respondents in the FCM group (73.4%) had a positive perception of care (score ≥35) than in the FS group (57.3%), p = 0.007 (Table 2).

On bivariate analyses, the factors significantly associated with perception of care in the FCM group was ethnicity, while it was ethnicity and study state in the FS group. Higher proportions of Igbo respondents compared to other ethnic groups had positive perception of care in both groups (p < 0.05). A significantly higher proportion of respondents from Kano state (69.2%) had positive perception of care in the FS group compared to respondents from Lagos (44.1%) (Table 3).

## Qualitative results

Four main themes were identified from qualitative analysis: perception of IV iron, experience with IV iron administration, comparison of IV and oral iron, and satisfaction with IV iron therapy.

**Theme 1: Perception of IV iron.** Although some women were initially scared or doubtful of an unknown treatment, there was generally a positive perception about IV iron. It was perceived to be good, effective, convenient, and fast-acting. A reason for positive perception about IV iron was the information relayed by providers, which was perceived to allay fears or anxiety related to the treatment. Another reason was the immediate improvement in their anaemia symptoms after receiving IV iron. Several women also noted that IV iron was advantageous as it reduced the need for blood transfusion (Table 4).

**Theme 2: Experience with IV iron administration.** Most women reported having a positive experience with IV iron administration, highlighting the quality of care received from providers. Most reported not having side effects or other challenges and some women had already recommended the treatment to others. However, a few women reported provider challenges with difficult intravenous access and with side effects. The challenges included repeated skin piercing, pain and injection phobia. Regarding the experience of side effects, one woman reported "not feeling good" on the day she received the administration. Another woman reported having a transient headache, while two women reported having a fever the day after administration; another woman required administration of another drug to counter side effects. Nevertheless, the women who reported challenges felt the treatment worked well for them (Table 4).

**Theme 3: Comparison IV and oral iron.** Most women prefer IV to oral iron because it is given once, corrects anaemia faster and provides a better health outcome. Oral iron was said to be less preferred because of forgetfulness in taking the tablets and dislike of taking tablets during pregnancy or generally. One woman complained about the smell of oral iron, which makes her prefer IV iron. Just a few women preferred oral iron to IV iron; two women who had relayed challenges with difficult intravenous access said they preferred oral iron because they didn't like their skin to be pierced. Another woman who had not used oral iron before, indicated interest in trying oral iron instead of IV in a future pregnancy (Table 5).

**Theme 4: Satisfaction with IV iron.** Overall, all the women who received IV iron reported being satisfied with the treatment and were willing to use it in subsequent pregnancies, including most of those who reported challenges. They expressed reasons for their satisfaction such as the quality of care received from providers, positive health outcome, no cost for the treatment and no need to continuously take oral iron (Table 5).

**Table 2. Comparison of perception of care in the intravenous and oral iron groups.**

| Variables | Perception of care in FCM group (N = 128) n (%) | Perception of care in FS group (N = 124) n (%) | p-value |
|---|---|---|---|
| **Felt provider explained what had been done** | | | 0.122 |
| A few times | 2 (1.6) | 2 (2.4) | |
| Most times | 16 (12.5) | 27 (21.8) | |
| All the time | 110 (85.9) | 94 (75.8) | |
| **Felt provider spoke in a language they could understand** | | | 0.941 |
| Most times | 10 (7.8) | 10 (8.1) | |
| All the time | 118 (92.2) | 114 (91.9) | |
| **Felt provider explained why he/she was giving medicine** | | | 0.889 |
| A few times | 3 (2.3) | 3 (2.4) | |
| Most times | 8 (6.3) | 5 (4.0) | |
| All the time | 116 (90.6) | 115 (92.7) | |
| Missing | 1 (0.8) | 1 (0.8) | |
| **Felt she could ask provider any question** | | | 0.850 |
| Never | 11 (8.7) | 9 (7.3) | |
| A few times | 6 (4.7) | 6 (4.8) | |
| Most times | 14 (11.0) | 18 (14.50 | |
| All the time | 96 (75.6) | 91 (73.4) | |
| **Felt providers took the best care of her they could** | | | 0.559 |
| A few times | 1 (0.8) | 0 (0.0) | |
| Most times | 10 (7.8) | 8 (6.5) | |
| All the time | 117 (91.4) | 116 (93.6) | |
| **Ease of treatment administration** | | | **0.006** |
| Easy; no challenge | 106 (82.8) | 79 (63.7) | |
| Little challenge | 19 (14.8) | 35 (28.2) | |
| Moderate challenge | 3 (2.3) | 9 (7.3) | |
| Lot of challenge | 0 (0.0) | 1 (0.8) | |
| **Seriousness of side effects** | | | 0.164 |
| No challenge | 123 (96.1) | 114 (91.9) | |
| Little challenge | 2 (2.3) | 9 (7.3) | |
| Moderate challenge | 2 (1.6) | 1 (0.8) | |
| **Overall satisfaction with information and support received** | | | **0.005** |
| Fair | 1 (0.8) | 1 (0.8) | |
| Good | 24 (18.8) | 46 (37.1) | |
| Excellent | 103 (80.5) | 77 (62.1) | |
| **Would recommend treatment to a family member** | | | **0.018** |
| Very strongly do not recommend | 1 (0.8) | 2 (1.6) | |
| Strongly do not recommend | 0 (0.0) | 2 (1.6) | |
| Undecided | 1 (0.8) | 1 (0.8) | |
| Recommend | 4 (3.1) | 12 (9.7) | |
| Strongly recommend | 32 (25.0) | 45 (36.3) | |
| Very strongly recommend | 90 (70.3) | 62 (50.0) | |
| **Overall perception of care** | | | **0.007** |
| Negative (score <35) | 34 (26.6) | 53 (42.7) | |
| Positive (score ≥35) | 94 (73.4) | 71 (57.3) | |

**Table 3. Factors associated with perception of care in the intravenous and oral iron groups.**

| Variables | FCM group (N = 128) | | | FS group (N = 124) | | |
|---|---|---|---|---|---|---|
| | Negative | Positive | p-value | Negative | Positive | p-value |
| **Age group (years)** | | | 0.553 | | | 0.840 |
| <30 | 19 (24.7) | 58 (75.3) | | 36 (43.4) | 47 (56.6) | |
| ≥30 | 15 (29.4) | 36 (70.6) | | 17 (41.5) | 24 (58.5) | |
| **Marital status** | | | 0.183* | | | 0.717* |
| Single | 0 (0.0) | 6 (100) | | 1 (100) | 0 (0.0) | |
| Cohabiting | 0 (0.0) | 2 (100) | | 1 (50.0) | 1 (50.0) | |
| Married | 33 (27.7) | 86 (72.3) | | 51 (42.2) | 70 (57.9) | |
| Divorced | 1 (0.0) | 0 (0.0) | | 0 (0.0) | 0 (0.0) | |
| **Education** | | | 0.821* | | | 0.508 |
| No formal education | 2 (22.2) | 7 (77.8) | | 1 (20.0) | 4 (80.0) | |
| Completed primary | 4 (30.8) | 9 (69.2) | | 9 (47.4) | 10 (52.6) | |
| Completed secondary | 19 (29.7) | 45 (70.3) | | 23 (37.7) | 38 (62.3) | |
| Completed tertiary | 8 (20.5) | 31 (79.5) | | 19 (52.8) | 17 (47.2) | |
| Postgraduate | 1 (33.3) | 2 (66.7) | | 1 (33.3) | 2 (66.7) | |
| **Place of residence** | | | 0.437* | | | 0.459* |
| Rural | 3 (37.5) | 5 (62.5) | | 4 (57.1) | 3 (42.9) | |
| Urban | 31 (25.8) | 89 (74.2) | | 49 (41.9) | 68 (58.1) | |
| **Ethnicity** | | | **0.028*** | | | **<0.001*** |
| Hausa | 13 (20.3) | 51 (79.7) | | 18 (29.0) | 44 (71.0) | |
| Igbo | 1 (7.14) | 13 (92.9) | | 3 (20.0) | 12 (80.0) | |
| Yoruba | 16 (39.0) | 25 (61.0) | | 28 (75.7) | 9 (24.3) | |
| Others | 4 (44.4) | 5 (55.6) | | 4 (40.0) | 6 (60.0) | |
| **Occupation** | | | 0.842* | | | 0.701* |
| Unemployed | 14 (27.5) | 37 (72.5) | | 23 (46.9) | 26 (53.1) | |
| Self-employed | 17 (27.4) | 45 (72.6) | | 18 (36.0) | 32 (64.0) | |
| Private employment | 3 (33.3) | 6 (66.7) | | 8 (53.3) | 7 (46.7) | |
| Government employment | 0 (0.0) | 4 (100) | | 3 (37.5) | 5 (62.5) | |
| Unknown/ Not applicable | 0 (0.0) | 2 (100) | | 1 (50.0) | 1 (50.0) | |
| **State** | | | 0.128 | | | **0.005** |
| Kano | 14 (20.9) | 53 (79.1) | | 20 (30.8) | 45 (69.2) | |
| Lagos | 20 (32.8) | 41 (67.2) | | 33 (55.9) | 26 (44.1) | |
| **Facility type** | | | 0.123* | | | 0.663* |
| Primary | 15 (38.5) | 24 (61.5) | | 16 (41.0) | 23 (59.0) | |
| Secondary | 17 (22.1) | 60 (77.9) | | 34 (45.3) | 41 (54.7) | |
| Tertiary | 2 (16.7) | 10 (83.3) | | 3 (30.0) | 7 (70.0) | |

*Fisher's exact p-value

## Data integration

Integration of key quantitative and qualitative results showed high convergence as illustrated in Table 6.

## Discussion

This mixed-methods study examined pregnant women's perceptions, experiences, and satisfaction with intravenous iron (FCM) compared to oral iron (FS) within the IVON type 1 hybrid effectiveness–implementation trial. By focusing on

**Table 4. Illustrative quotes for themes 1 and 2.**

| Theme 1 – Perception of IV iron |
| --- |
| *"It's good because so far I have seen improvement in my blood level…I was given infusion - the drip - last month and after two weeks that I came around my blood has improved, so it's good so far"* [31-35 years, secondary facility, Lagos] |
| *"My belief about it, it's good, it's good and effective and I think at first I was scared to have it but at the long run after having it, it's fine. It's very effective and convenient to me because that is just once rather than taking tablets daily, sometimes you might forget to take your tablet so that's more effective and convenient."* [36-40 years, secondary facility, Lagos] |
| *"Yes, at first I was scared, but since I was given the necessary information, the fear went out of my mind and I saw how I was being taken care of, so I came back to normal."* [16-20 years, PHC, Kano] |
| *"Hmm, what I feel is that, to me it's better than the other one* [oral iron] *because they say I shouldn't use any blood* [transfusion] *…I tried to work towards it but what I just have in mind is that when it gets closer to my delivery time, if I see that my blood …is still low, that what I will do is that I will go and buy another drug, another blood drug to be using, but so far I have been coming, it's* [blood level] *increasing, it's increasing apart from I am eating vegetables. So, I feel it's much better."* [31-35 years, tertiary facility, Lagos] |
| *"I became stronger and after I went back home … honestly I am very grateful…all those discomforts disappeared; no body swelling, there wasn't shortage of breath anymore, the body weakness and dizziness disappeared, honestly after the IV injection all the aforementioned disappeared."* [21-25 years, PHC, Kano] |
| *"What I have to say is that if someone has low blood in pregnancy, I prefer them going into this Ivon [IV iron] than telling them to go and buy blood, this one is okay than using other's blood…I think this Ivon [IV iron] is okay than that."* [21-25 years, secondary facility, Lagos] |
| Theme 2 – Experience with IV iron administration |
| *"Honestly there has not been any challenge. What challenge would there be with all that quality care they gave me, even after the drip administration I was asked to stay back for like 30 minutes, and they kept monitoring my health throughout. I am very grateful for that."* [31-35 years, tertiary facility, Kano] |
| *"Eeeh! My experience it was a pleasant one the apart from the injection ok was very OK. The doctor, yeah cause I have a very small vein it was very difficult to get the vein but apart from that the customer should I say the customer care was something else, the doctor care rather was exceptional."* [26-30 years, secondary facility, Lagos] |
| *"I have even told people that the treatment is very good and I am not the only one that took the drip because I see when the nurse will call other pregnant women and start doing tests for them. So, the treatment is very good, and it works very well and I have told lots of people about it."* [31-35 years, PHC, Lagos] |
| *"It is very easy to take except the part where they need to put needle in my hand and you know I am afraid of the needle. But it is very good because it worked fast and I was strong after taking the drip."* [21-25 years, PHC, Lagos] |
| *"The bad part was what I said, I only felt somehow when I first took the drip, I wasn't really feeling good. That is the only challenge I faced but based on the drug, I felt so well. It was only the drip that didn't go well with me the first day I took it."* [36-40 years, tertiary facility, Lagos] |
| *"Ok at the initial I was having a headache which she said was going to subside and that was the only thing I had…. I had slight headache which she said I should rest but when I got home it was as if I didn't get anything."* [26-30 years, secondary facility, Lagos] |
| *"Haa it was like all over my body was vibrating and the thing turn me upside down. So, the thing do me something (Laughing) I can't even explain that one. So later I went to the clinic and the nurse gave me another drug and all the things disturbing my body calmed down….I know that the drip is very good and special for pregnant women that have shortage of blood. So, I know it did positively well in my body."* [21-25 years, PHC, Lagos] |

client-reported outcomes, this paper complements earlier IVON publications on effectiveness [18] and implementation outcomes, including acceptability [10] and fidelity [28]. Together, these studies provide a multi-layered understanding of IV iron introduction in Nigeria through the Proctor implementation outcomes framework. [15]

Our findings demonstrate that most women receiving FCM reported positive experiences, rating care as easy and satisfactory, and describing rapid symptomatic improvement and convenience. More of the women in the FCM group had overall positive perception of treatment compared to those in the FS group. Ethnicity and state of residence were factors associated with perception of treatment; these may be due to a high level of complementariness in language and/ or ethnicity of providers delivering care at those facilities. Qualitative narratives revealed that single-dose administration, minimal side effects, and effective provider communication fostered confidence and trust. These perceptions reflect not only intrinsic features of the intervention such as reduced pill burden and rapid correction of anaemia, but also contextual factors: counselling quality, close trial follow-up, and the absence of financial cost. Hence, participants' favourable views of IV iron likely represent an interaction between intervention characteristics and supportive trial conditions, rather than

**Table 5. Illustrative quotes for themes 3 and 4.**

| Theme 3 – Comparison IV and oral iron |
| --- |
| "I think it's better yeah for me it's better and for most treatment it's better. Why? Because any pregnancy you know the hormones system tend to have so many funny attitude so for pregnant women I don't think we have that time to start taking pills every day. Some people don't even like pills. I don't like it; I prefer injections to pills, so for someone like me that does not like a lot of drama I take something once and it serves you throughout the pregnancy. I think it makes sense. I think it just makes sense. I can remember…during my first pregnancy I missed out taking my routine drugs a lot of time so if I can just take something once and it is going to take me throughout the pregnancy I think it's just best." [26-30 years, secondary facility, Lagos] |
| "Hmm, I have never used the pills before because this is my first pregnancy. So, but with the way this one worked for me; because during my delivery with CS [Caesarean section], I didn't collect blood [for emergency use]. They collected 1 pint of [donor] blood because it's emergency CS…but we didn't use it. It was later returned. So, I find it very okay more than pills… So, I see that this one is more active than the pill." [31-35 years, tertiary facility, Lagos] |
| "Because…like I said it's a one-off something, once it's taken it's taken, I don't have to start taking it in the morning … I don't have to take it again all through the pregnancy. That's why I prefer it …I never noticed any nothing like allergy or abnormality in me at least, it's okay." [31-35 years, tertiary facility, Lagos] |
| "Honestly, I became much healthier with this pregnancy after I was given the IV iron than other pregnancies that I took the oral iron pills. Both work in a gradual process but I was much healthier after the IV iron administration than when I was taking the iron pills." [21-25 years, PHC, Kano] |
| "When you talk about tablet, I am not someone that likes to take tablets but when they did the test for me and I said I will take the drip… I was happy that it is once I will take the drip and it's not the tablets that I will be taking three times daily. You know tablets is something someone can forget to take in the morning, afternoon or evening and someone can say I don't want to use the tablet today jare (Hisses)." [26-30 years, PHC, Lagos] |
| "It [oral iron] doesn't pain, you don't need, they don't need to chook [pierce] you anything, you just swallow it, but drip they have to pierce your hand that's it…I don't like it." [21-25 years, secondary facility, Lagos] |
| **Theme 4 – Satisfaction with IV iron** |
| "I am very satisfied with it, I can't thank the team enough because they diagnosed me, enlightened me about it, they gave me treatment and kept monitoring my progress." [31-35 years, tertiary facility, Kano] |
| "Em Haa! I'm really satisfied cos even after I gave birth I did not experience any dizziness which I experienced during my first child in form of dizziness or weakness or anything. All those funny abnormalities that I experienced before the delivery went off, so they are not there anymore so I'm a happy person." [26-30 years, secondary facility, Lagos] |
| "I am very happy with the treatment because all the things that was doing me before, stopped after I received the treatment and my body was back to normal and there were lots of good changes in my body till and after I gave birth. … Ahh the nurse always calls me on phone if I am supposed to come to the clinic that day and I am not there. So, she will call me that they are expecting me and I will come. So, I am very happy because after receiving the drip my body was very strong and my baby too was strong. So, as you can see my baby is 41 days now and you can see how strong and happy he is." [31-35 years, PHC, Lagos] |
| "I am very happy because if you want to go do blood test outside, you will need to pay some money, but it has been done free of charge for us here. Also, the drip that was given to me was free, assuming I did the blood test outside with money, they will also give me the drip after I pay some money. Even if they say I should buy blood tonic, I will need to buy it with money. So, I am very happy that all this was done for me free here… You know anytime I come to the clinic, nurse will always do another check-up for us and if there is any complaint and she doesn't understand, she will call the doctor. So, doctor will tell her to do so and so things for us and she will do it. So the nurse was very helpful and she always calls us on phone if we have to come to the clinic, so I am very happy and satisfied with the treatment." [26-30 years, PHC, Lagos] |
| "I really enjoyed it…..I know that I would readily accept IV iron administration when next I get anaemia in pregnancy." [21-25 years, secondary facility, Kano] |

intrinsic superiority of FCM alone. This distinction is critical when interpreting client satisfaction as an outcome within a controlled setting.

The strong emphasis on communication and empathy in participant narratives underscores the role of provider–patient relationships in influencing treatment acceptance. Similar findings in other LMIC contexts show that trust, perceived attentiveness, and reassurance by providers enhance uptake of new maternal interventions [29,30]. In this regard, the high satisfaction observed may partly signify successful trial implementation, characterized by effective education, careful monitoring, and responsiveness to concerns, all of which are essential precursors for real-world adoption. Also, considerations should be made for diverse ethnicities and cultures in real-world adoption; first, in exploring how these demographic factors influence perception of care and second, in the delivery of culturally sensitive care [31].

**Table 6. Illustrating the integration of quantitative and qualitative findings.**

| Key themes | Quantitative findings | Qualitative findings | Examples of representative quotations | Interpretation of integration |
|---|---|---|---|---|
| Perception of care with IV iron | 90.6% of participants felt that the provider explained the reasons for administering the medication. | Overall, there was a positive perception of IV iron treatment. Participants indicated that the information shared by healthcare providers helped to alleviate their fears or anxieties regarding the treatment. | *"…I can't thank the team enough because they diagnosed me, enlightened me about it, they gave me treatment and kept monitoring my progress."* [31-35 years, tertiary facility, Kano] | High convergence: Most participants were informed about the intervention and its rationale, which aligns with the generally positive feedback received. |
| Experience with IV iron administration | 82.8% of participants reported ease of treatment administration, while 14.8% experienced minimal challenges, and only a small percentage (1.6%) faced moderate challenges. | Most reported not having challenges, and some women had already recommended the treatment to others. However, a few women reported challenges with setting the IV line. | *"Eeeh! My experience it was a pleasant one the apart from the injection ok was very OK. The doctor, yeah cause I have a very small vein it was very difficult to get the vein but apart from that the customer should I say the customer care was something else, the doctor care rather was exceptional."* [26-30 years, secondary facility, Lagos] | High convergence: Most participants reported no challenges during the administration process, which is consistent with the overall positive feedback. Additionally, those who faced challenges still felt that the treatment was effective for them. |
| Experience with IV iron administration | 96.1% of participants had no side effects during or after the administration of IV iron therapy. | The narratives from the qualitative findings showed that most participants reported no side effects from the IV iron therapy. However, a few women reported challenges with side effects. | *"Ok at the initial I was having a headache which she said was going to subside and that was the only thing I had…. I had slightly headache which she said I should rest but when I got home it was as if I didn't get anything."* [26-30 years, secondary facility, Lagos] | High convergence: The survey aligns with the qualitative data, showing that most participants reported no side effects. Additionally, the findings include details on how the few participants who experienced side effects were managed. |
| Satisfaction with IV iron | 80.5% of all participants reported excellent satisfaction with the information and support they received. | Overall, all women who received IV iron expressed satisfaction with the treatment and indicated a willingness to use it in future pregnancies, even among those who faced challenges. | *"I really enjoyed it…. I know that I would readily accept IV iron administration when next I get anaemia in pregnancy."* [21-25 years, Secondary Facility, Kano] | High convergence: The survey results reveal that the majority of participants had an excellent overall level of satisfaction, which corresponds with the qualitative feedback indicating that almost all women were pleased with the intervention and were inclined to accept it in subsequent pregnancies. |

From an implementation science perspective, the convergence of high fidelity, high acceptability, and positive client satisfaction across IVON sub-studies indicates that IV iron introduction was feasible and well-received when delivered with adequate system support. These complementary findings, which include clinical effectiveness as a service outcome, adherence to protocol as an implementation outcome, and positive user experience as a client outcome (Fig 3), collectively strengthen the evidence base for potential scale-up. However, integrating IV iron into routine antenatal care will require careful attention to contextual constraints such as supply-chain reliability, provider training, and affordability for pregnant women outside the trial environment.

**Strengths and limitations**

The study's strengths lie in its robust mixed-methods design, integration of quantitative and qualitative data, and grounding in an implementation research framework. Limitations include possible recall bias, particularly for FCM, as participants were interviewed several weeks after treatment, and social desirability bias within a supervised trial setting. Consequently, perceptions may differ under routine service conditions. Furthermore, purposive sampling in the qualitative arm, while enriching depth, limits generalizability to the broader population of pregnant women in Nigeria.

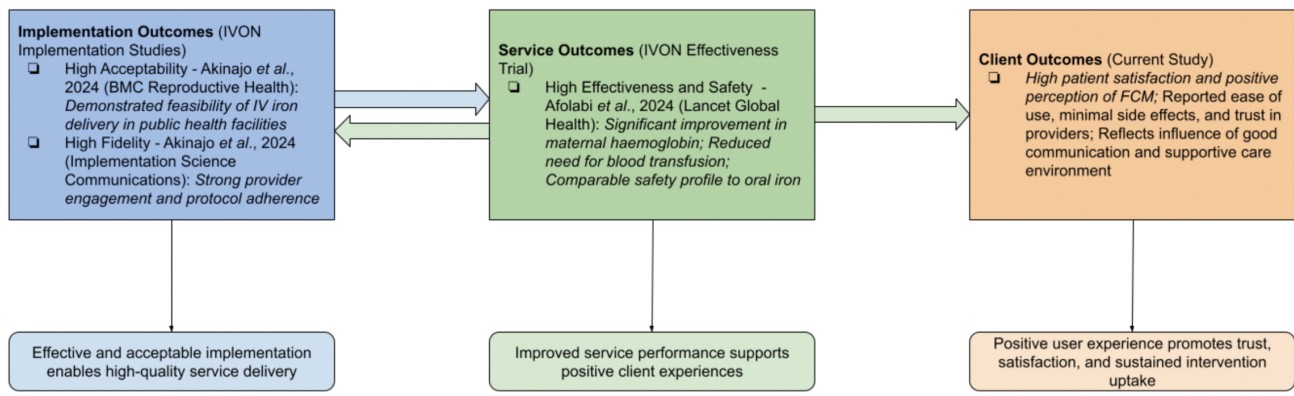

*Together, these findings illustrate the layered progression from implementation and service performance to patient-level satisfaction within the Type 1 hybrid effectiveness–implementation design of the IVON trial.*

**Fig 3. Integration of IVON outcomes using Proctor's (2011) implementation outcomes framework.**

## Conclusion

This study adds a critical patient-centred dimension to the IVON research portfolio, documenting positive perceptions and high satisfaction with IV iron therapy among Nigerian pregnant women. When interpreted alongside previously published evidence of high effectiveness, fidelity, and acceptability, these findings suggest that IV iron is both feasible and well-received within supportive health-system conditions. Nevertheless, because this work was conducted in a controlled trial environment, broader adoption will depend on addressing affordability, health-system capacity, and sustained provider engagement. Future implementation studies should evaluate these client outcomes under real-world programmatic conditions, incorporating cost-effectiveness and equity considerations to guide national policy on anaemia management in pregnancy.

## Supporting information

**S1 Checklist. Standards for Reporting Implementation Studies (StaRI) checklist.**
(DOCX)

## Acknowledgments

We wish to acknowledge the IVON trial study team in Lagos and Kano states for their support for this study. We appreciate the IVON research assistants (Binta Umar Abdullahi, Amina Abubakar, Rukayya Sadiqa Isa, Judith Amos, Jennifer Ejiofor, Oluchi Ozonu, Jacob Igologba, Kehinde Adeshina, Donald Ezinwanne) for their hard work in collecting data. Finally, we are deeply grateful to the postpartum women who took the time to share their experiences for this study.

## Author contributions

**Conceptualization:** Mobolanle Balogun, Opeyemi R. Akinajo, Nadia A. Sam-Agudu, Bosede B. Afolabi.

**Data curation:** Hameed Adelabu.

**Formal analysis:** Mobolanle Balogun, Rachel A. Thompson.

**Funding acquisition:** Bosede B. Afolabi.

**Investigation:** Mobolanle Balogun, Opeyemi R. Akinajo.

**Methodology:** Mobolanle Balogun, Opeyemi R. Akinajo, Nadia A. Sam-Agudu, Bosede B. Afolabi.

**Project administration:** Rachel A. Thompson, Teniola Lawanson.

**Supervision:** Nadia A. Sam-Agudu, Bosede B. Afolabi.

**Writing – original draft:** Mobolanle Balogun, Opeyemi R. Akinajo, Rachel A. Thompson, Teniola Lawanson, Hameed Adelabu.

**Writing – review & editing:** Nadia A. Sam-Agudu, Bosede B. Afolabi.

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
