## [Decision Letter · Decision Letter 0]

30 Jul 2025

Dear Dr. Balogun,

**Please respond to each comment from both reviewers.**

We look forward to receiving your revised manuscript.

Kind regards,

Jennifer Yourkavitch

Academic Editor

PLOS ONE

**Journal Requirements:**

1. When submitting your revision, we need you to address these additional requirements. Please ensure that your manuscript meets PLOS ONE's style requirements, including those for file naming. The PLOS ONE style templates can be found at https://journals.plos.org/plosone/s/file?id=wjVg/PLOSOne_formatting_sample_main_body.pdf and https://journals.plos.org/plosone/s/file?id=ba62/PLOSOne_formatting_sample_title_authors_affiliations.pdf 2. Thank you for stating the following financial disclosure: The main trial titled “Intravenous versus oral iron for iron deficiency anaemia in pregnant Nigerian women (IVON)” is an open-label, randomised controlled trial funded by the Bill & Melinda Gates Foundation (BMGF) Grant (Investment ID INV-017271).    Please state what role the funders took in the study.  If the funders had no role, please state: "The funders had no role in study design, data collection and analysis, decision to publish, or preparation of the manuscript." If this statement is not correct you must amend it as needed. Please include this amended Role of Funder statement in your cover letter; we will change the online submission form on your behalf. 3. If the reviewer comments include a recommendation to cite specific previously published works, please review and evaluate these publications to determine whether they are relevant and should be cited. There is no requirement to cite these works unless the editor has indicated otherwise. 

Reviewers' comments:

**Comments to the Author**

1. Is the manuscript technically sound, and do the data support the conclusions?

Reviewer #1: Partly

Reviewer #2: Partly

2. Has the statistical analysis been performed appropriately and rigorously?

Reviewer #1: I Don't Know

Reviewer #2: No

3. Have the authors made all data underlying the findings in their manuscript fully available?

Reviewer #1: No

Reviewer #2: No

4. Is the manuscript presented in an intelligible fashion and written in standard English?

Reviewer #1: Yes

Reviewer #2: Yes

**Reviewer #1:**  While I find the study important, I am a bit concerned about the Methodology and interpretation of the findings.

The title is misleading in that it does not indicate what is actually assessed. The current title: Patient outcomes in a Type 1 hybrid implementation research on intravenous iron for

anaemia in pregnancy in Nigeria: a mixed methods study - does not indicate that this is an assessment of perception of the treatment. It rather indicates that it is on treatment outcomes, which it is clearly not.

The selection process for participants is not explained - how were participants chosen? It is not entirely clear how many participants were selected and hwo many dropped out - a flowchart would have been appropriate.

The interpretation of the positive perception is likely a reflection of how the protocol was implemented in this study, but it is not clear how well this would be done outside of a study setting. It would have been good to ask if they would have paid for the treatment since cost seems to be an important factor for this delivery method.

The qualitative results seem quite selective - it may be good to analyze those some more and group the answers on common themes.

In general, I find the study interesting but in my opinion it will require a significant effort to make it publishable.

**Reviewer #2:**  While my comments may appear critical, i would like to note that I think your work is commendable. Thank you. I tried to engage with it deeply in my attempt to increase clarity. I would welcome future revisions, if you choose to consider my comments. Please see attached document with comments.

**Do you want your identity to be public for this peer review?** For information about this choice, including consent withdrawal, please see our Privacy Policy

Reviewer #1: **Yes:** Laura S. Hackl

Reviewer #2: **Yes:** Denish Moorthy

---

## [Author Response · Author response to Decision Letter 1]

4 Nov 2025

Point-by-point reviewers' comments and authors' answers

Title of the manuscript: Patients’ perceptions of care in a Type 1 hybrid effectiveness-implementation trial on intravenous iron for anaemia in pregnancy in Nigeria’

Response to reviewer's comments:

No Reviewers' comments Authors' response

Reviewer 1

1. The title is misleading in that it does not indicate what is actually assessed. The current title: Patient outcomes in a Type 1 hybrid implementation research on intravenous iron for anaemia in pregnancy in Nigeria: a mixed methods study - does not indicate that this is an assessment of perception of the treatment. It rather indicates that it is on treatment outcomes, which it is clearly not. Thank you for your comments and generally for your review which has helped to improve our paper. Patient outcomes refer to measures of wellbeing that encompass mental health, quality of life, perception of and satisfaction with care, and physical health, as well as the use of services and treatments. However, we have taken on board your title suggestion and that of reviewer 2 and rephrased the title as follows: Patients’ perceptions of care in a Type 1 hybrid effectiveness-implementation trial on intravenous iron for anaemia in pregnancy in Nigeria.

2. The selection process for participants is not explained - how were participants chosen? It is not entirely clear how many participants were selected and how many dropped out - a flowchart would have been appropriate.

Thank you for your comments. The selection process was explained in the manuscript precisely under the data collection methods for the quantitative aspect of the study, as “Of the 264 participants contacted across all the healthcare facilities, 260 were reachable, and 252 agreed to participate in the exit surveys.” (page 7, lines 172 – 174). For the qualitative aspect, it was stated as “In total, we interviewed 66 participants, purposively selected from all eleven healthcare facilities concurrently with the exit surveys.” (page 8, lines 183 – 184).

For clarity and understanding for the readers, we have included a flowchart as a figure, as suggested, which is referred to as Figure 2 in the manuscript.

3. The interpretation of the positive perception is likely a reflection of how the protocol was implemented in this study, but it is not clear how well this would be done outside of a study setting. It would have been good to ask if they would have paid for the treatment since cost seems to be an important factor for this delivery method. Thank you for your comment. For the survey results, we have now rephrased the write-up to replace “perception of treatment” with “perception of care” , which is influenced by both the technical quality of care and the interpersonal aspects of the patient experience, and is more reflective of what we assessed. We also appreciate that the positive perception could have been influenced by the controlled trial setting and may not be reflective of real-world situation and we had noted this as a study limitation. We also clarified in our discussion that participants’ satisfaction likely reflects trial implementation quality (effective counselling, close monitoring, free treatment) rather than intrinsic superiority of IV iron.

We did not ask about willingness to pay as that was outside the scope of our study. We noted as part of our recommendations in the conclusion that “Future implementation studies should evaluate these client outcomes under real-world programmatic conditions, incorporating cost-effectiveness and equity considerations to guide national policy on anaemia management in pregnancy.”

4. The qualitative results seem quite selective - it may be good to analyze those some more and group the answers on common themes. Thank you for your suggestion. We went over the transcripts again and arrived at the same themes. This is because we used a deductive approach for our coding based on our indepth interview guide. The guide directly addresses our research questions relating experience and satisfaction with IV iron.

Reviewer 2

General comment: I would suggest to the authors that the qualitative results could be presented without drawing broad conclusions from the analysis. Some of the conclusions would require presentation of implementation and service outcomes too and triangulating those with client outcome. I presume will be published in other papers – that said, I think that triangulation of service, implementation and client outcomes together would be a more valuable paper, as I note in general comments below. I have outlined general issues below, followed by specific comments on the various sections Thank you for your review that has improved our paper considerably. As you already imagined, the service and implementation outcomes of this work have already been published. However, we have triangulated the key findings from the published sub studies with this one in our discussion. We have rewritten the discussion section and removed instances where we have drawn broad conclusions from our qualitative results.

1. Language: I found the text to be unnecessarily complicated in its explanation. For example, in line 100, “This mixed-methods study employs a convergent parallel design, combining quantitative and qualitative data collection methods to provide a comprehensive perspective on the users' perceptions, experience and satisfaction with IV iron therapy” – an example of language that could be simplified.

Thank you for your comment. We have rephrased this sentence as suggested: “This mixed-methods study combines quantitative and qualitative data collection methods to provide a comprehensive perspective on the users' perceptions, experience and satisfaction with IV iron therapy.” (page 4, lines 102-104). We have also simplified language in other areas of the manuscript.

2. Two of the limitations noted in the paper – controlled trial setting and recall bias - also serve to highlight that the findings specifically mentioned in this paper are: (i) not representative of the population or the real world setting.; (ii) provide a positive impression of an intervention that has occurred in the past (FCM) against a intervention that is ongoing (oral iron; presumably ongoing, since the women are still pregnant). I’d suggest explicitly considering that when weighing the differences between the interventions. Thank you for these points. We have rewritten the discussion and clarified that participants’ satisfaction likely reflects trial implementation quality (effective counselling, close monitoring, free treatment) rather than intrinsic superiority of IV iron.

We agree that there are clear differences between the two interventions. FCM given once versus months of treatment for oral iron (this information is now included in the paper in lines 188-190) are intrinsic features of treatment that are non-modifiable and could influence perception of end-users. The single dose was perceived to be advantageous in the qualitative study and we have noted this in the discussion.

3. Introduction could use more information on the rationale behind the switch from oral to IV iron (noted in line 65-66) and how that informs the design of the tools in this current study. Thank you for your comment. Lines 98 – 108 detail the rationale behind the switch from oral to IV iron while the justification for our study is presented in lines 124-134.

4. In my reading, I felt that the authors left many issues unexplored in the quant and qual analysis, which focused only on numbers and percentages – some of the deeper analysis could include association with background characteristics, facilities where they were seen and why, tertiary versus secondary versus primary health care facility. There is an opportunity to go deeper into their health care seeking beliefs and behaviors, and why the mothers feel that way about IV iron in that context, but the analysis remains superficial Thank you for your suggestion. We have now done further analysis including association of sociodemographic variables with overall perception, which is presented in Table 3. For the qualitative data, we went over the transcripts again and arrived at the same themes. This is because we used a deductive approach for our coding based on our indepth interview guide. The guide directly addresses our research questions relating experience and satisfaction with IV iron.

5. Rather than a vote counting of how many mothers in each group rated which service, I would suggest a deeper exploration of why the pregnant women said what they said. However, the themes on IV iron – perception, experience , comparison with oral and satisfaction – are superficial and reflect the rationale of why we are using IV as an intervention, i.e. as a response to the challenges that mothers have faced with oral iron and its use. Restating those reasons in the context of a study is always useful, but this is neither new nor innovative and does not move the needle on adoption of IV iron. The practice of presentation of numbers and “vote counting” in the context of the paper is not very useful and subject to bias, as the sampling was purposive. The differences are not meaningful as they could be due to differences in the mothers who were selected into each group using pre-identified criteria Thank you for your valid comments. We would like to clarify that the participants for the exit surveys were not selected purposively but through systematic random sampling. We used a sampling interval of 4 to select 25% of women and there was no predetermined criteria for their selection. We have made this clearer in the paper: “We aimed to survey 25% of women from each study arm, using systematic random sampling programmed into the data collection platform, to select every fourth participant, resulting in a total of 132 participants in each group and 264 in total”. (page 6, lines 216 – 218).

It was the IDI participants that were purposively selected based on certain characteristics, as described in the manuscript (Page 6, lines 221-226).

The sampling procedure has been further illustrated in Figure 2 as suggested.

6. As stated, the results gives the reader the impression that one intervention is better than the other, while in fact, it is documenting the perception of pregnant mother vis-a-vis her interaction with the health provider. As it stands, FCM is easy because its ease of administration is a feature, explicitly designed to counter the oft-repeated complaint of the challenges of taking pills throughout pregnancy. I interpret that as a confirmation of the design of the intervention and not reflective of the differences between the interventions Thank you for catching this. We have rephrased the overall satisfaction written in lines 304-305 to relate to “satisfaction with information and support received from providers” as that was how the question was phrased.

We believe that clients’ perception of the design of an intervention is also relevant. We did not want to assume, for example, that FCM is easy to administer for all women. Some participants in the qualitative study reported dislike for skin piercing, pain and injection phobia.

To address your concern, we have rewritten the discussion section and clarified that participants’ satisfaction likely reflects trial implementation quality rather than intrinsic superiority of IV iron.

7. Incomplete picture of Type 1 hybrid implementation research: The full picture of implementation should include all outcomes – service, implementation, and client. As a stand-alone paper, while these findings are useful to document, they are neither comprehensive nor insightful. They would be valuable when correlated and presented together with implementation and service outcomes Thank you for your valid points. Since the service and implementation outcomes of this work have already been published, we have now correlated the key findings from the published sub studies with this one and presented them together in an implementation research framework (figure 3) in our discussion.

8. Title: I found the title promising but felt that the paper content did not match its promise, as per my comments above. The research framework only features in the introduction, and mixed methods design has multiple issues in interpretation. I’d suggest a simpler descriptive title. Thank you for your comment. We have taken on board your title suggestion and that of reviewer 1 and rephrased the title as follows: Patients’ perceptions of care in a Type 1 hybrid effectiveness-implementation trial on intravenous iron for anaemia in pregnancy in Nigeria.

Specific comments

Abstract

1. Results section – suggest rewriting this section to include more qualitative findings; the quantitative findings are more descriptive than informative Thank you for your comments. We have rewritten the results section to include more qualitative findings.

2. Line 44 – it is not the treatment that is rated as excellent (as per table 2); it is the information and services provided around the treatment. Since this is a trial, this speaks more to the deficiencies in the oral iron arm rather than the benefits of the IV iron arm. Thank you for catching this. This has been deleted here and replaced with result on overall perception of care and it has rephrased in the main paper.

3. Line 49: I would not state it confidently that this study has established IV iron as a “viable alternative to oral iron”. Consider being more circumspect. Thank you for this valid comment. This has now been rephrased in the conclusion

Introduction

1. Line 56 – reconsider using AIP as an acronym; it is not a long phrase and AIP is not a standard acronym Thank you for your suggestion. Anaemia in pregnancy has been written in full throughout the paper.

Methods

1. Line 100-105: See general comments. Thank you. We have simplified the language in this section.

2. Line 138 – Please clarify how these participants were selected; “equates to” is not the same as "we selected every fourth participant.

Thank you for the observation. We have made the statement clearer as suggested: “We aimed to survey 25% of women from each study arm, using systematic random sampling to select every fourth participant, resulting in a total of 132 participants in each group and 264 in total”. (page 6, lines 216 – 218)

3. Line 154 – is this the same as the exit survey? Its confusing -I read this to be another survey; please clarify at the beginning of the methods early on that the data sources are a 10 question exit survey and in-depth interviews Yes, it is the same survey that was referred to. For clarity, we have made the statement clearer: “For the exit survey, the research team designed a 10-item questionnaire to collect data on participant’s perceptions of care and satisfaction regarding the treatment (FCM or FS) they received during the trial.” (page 7, lines 241 – 243)

For the IDI, we have also rephrased as “Qualitative data were collected through in-depth interviews (IDIs) with the purposively selected participants. A semi-structured topic guide with five sections was developed to explore users' knowledge of anaemia in pregnancy, their perceptions of and relative advantages of IV iron compared to oral iron, and their challenges and experiences with the administration process, including overall satisfaction with IV iron therapy.” (page 7, lines 252 – 255)

4. Line 161 – Who reviewed it?

The research team reviewed the survey. We have made the statement clearer as “Before data collection, the survey underwent a rigorous review process by the research team, was corrected and was piloted with non-participating pregnant women to ensure the validity and reliability of the tool.” (page 7, lines 248 – 250)

5. Line 162 - validity and reliability of what exactly?

We were referring to the content of the exit survey in the statement. However, we have made it clearer as “Before data collection, the survey underwent a rigorous review process by the research team, was corrected and was piloted with non-participating pregnant women to ensure the validity and reliability of the tool.” (page 7, lines 248 – 250)

6. Line 163-164: unclear what the five sections explored? Same as the exit survey or differ

---

## [Decision Letter · Decision Letter 1]

2 Jan 2026

Patients’ perceptions of care in a Type 1 hybrid effectiveness-implementation trial on intravenous iron for anaemia in pregnancy in Nigeria

PONE-D-25-22404R1

Dear Dr. Balogun,

We’re pleased to inform you that your manuscript has been judged scientifically suitable for publication and will be formally accepted for publication once it meets all outstanding technical requirements.

Kind regards,

Jennifer Yourkavitch

Academic Editor

PLOS One

Reviewers' comments:

Reviewer's Responses to Questions

**Comments to the Author**

Reviewer #2: All comments have been addressed

2. Is the manuscript technically sound, and do the data support the conclusions?

Reviewer #2: Yes

3. Has the statistical analysis been performed appropriately and rigorously?

Reviewer #2: Yes

4. Have the authors made all data underlying the findings in their manuscript fully available?

Reviewer #2: Yes

5. Is the manuscript presented in an intelligible fashion and written in standard English?

Reviewer #2: Yes

Reviewer #2: All comments have been addressed. Thank you for thoughtfully considering the comments. Good luck with your research.

**Do you want your identity to be public for this peer review?** For information about this choice, including consent withdrawal, please see our Privacy Policy

Reviewer #2: **Yes:** DENISH MOORTHY

---

## [Editor Report · Acceptance letter]

PONE-D-25-22404R1

PLOS One

Dear Dr. Balogun,

I'm pleased to inform you that your manuscript has been deemed suitable for publication in PLOS One. Congratulations! Your manuscript is now being handed over to our production team.

Kind regards,

on behalf of

Dr. Jennifer Yourkavitch

Academic Editor

PLOS One